# Modulation of the Gut Microbiota with Prebiotics and Antimicrobial Agents from *Pleurotus ostreatus* Mushroom

**DOI:** 10.3390/foods12102010

**Published:** 2023-05-16

**Authors:** Gréta Törős, Hassan El-Ramady, József Prokisch, Fernando Velasco, Xhensila Llanaj, Duyen H. H. Nguyen, Ferenc Peles

**Affiliations:** 1Institute of Animal Science, Biotechnology and Nature Conservation, Faculty of Agricultural and Food Sciences and Environmental Management, University of Debrecen, Böszörményi Street 138, 4032 Debrecen, Hungary; 2Doctoral School of Animal Husbandry, University of Debrecen, Böszörményi Street 138, 4032 Debrecen, Hungary; 3Soil and Water Department, Faculty of Agriculture, Kafrelsheikh University, Kafr El-Sheikh 33516, Egypt; 4Doctoral School of Food Science, University of Debrecen, Böszörményi Street 138, 4032 Debrecen, Hungary; 5Tay Nguyen Institute for Scientific Research, Vietnam Academy of Science and Technology, Dalat 70072, Vietnam; 6Institute of Food Science, Faculty of Agricultural and Food Sciences and Environmental Management, University of Debrecen, Böszörményi Street 138, 4032 Debrecen, Hungary

**Keywords:** prebiotics, probiotics, *Pleurotus ostreatus* polysaccharides, phenolic compounds, terpenoids, lectins, oyster extracts, secondary metabolies

## Abstract

*Pleurotus ostreatus* (Jacq. ex Fr.) P. Kumm mushroom contains bioactive compounds with both antimicrobial and prebiotic properties, which are distributed in the mushroom mycelium, fruiting body, and spent substrate. The mushroom is rich in nondigestible carbohydrates like chitin and glucan, which act as prebiotics and support the growth and activity of beneficial gut bacteria, thereby maintaining a healthy balance of gut microbiota and reducing the risk of antibiotic resistance. The bioactive compounds in *P. ostreatus* mushrooms, including polysaccharides (glucans, chitin) and secondary metabolites (phenolic compounds, terpenoids, and lectins), exhibit antibacterial, antiviral, and antifungal activities. When mushrooms are consumed, these compounds can help preventing the growth and spread of harmful bacteria in the gut, reducing the risk of infections and the development of antibiotic resistance. Nonetheless, further research is necessary to determine the efficacy of *P. ostreatus* against different pathogens and to fully comprehend its prebiotic and antimicrobial properties. Overall, consuming a diet rich in mushroom-based foods can have a positive impact on human digestion health. A mushroom-based diet can support a healthy gut microbiome and reduce the need for antibiotics.

## 1. Introduction

Champignon mushroom (*Agaricus bisporus*) is the first cultivated and distributed edible mushroom worldwide, closely followed by *Pleurotus* genus [1]. The oyster mushroom (*Pleurotus ostreatus*) has great nutritional and medicinal attributes, which make this mushroom to be the second most popular edible and cultivated mushroom in the world after *Agaricus bisporus* [2]. Among all the several health benefits reported by many numerous published studies [3,4], this mushroom has also shown promising potential for green biotechnology [1,5], biorefinery, and recycling of agro-wastes [6], as well as pharmacological potential and medicinal properties [2,7]. In recent years, there has been a growing interest in extracts and the isolated compounds present in *P. ostreatus* mushrooms due to their potent antibacterial [2,8,9], antifungal [8], antiviral [2,10,11,12,13,14], antihelmintic [15,16], and pharmacological effects against different microorganisms. Furthermore, there is interest in their potential growth stimulation activity of probiotic bacteria [17,18]. Many of these bioactive compounds play a potential tool in several diseases’ treatment and gut microbiota modulation, so they are considered to be functional food and medicine ingredients. The literature data indicate that the *P. ostreatus* mushroom contains a high range of phytochemicals with antimicrobial and prebiotic functions, like alkaloids, steroids, flavonoids, saponins, tannins, phenolic components [19], glycosides, and terpenoids [20,21]. Interestingly, a positive correlation has been found between phenols and the regulatory effect of probiotics, which results in the promotion of a healthy balance of microorganisms in the gut [22].

The gut microbiota comprises different communities of microorganisms (>1000 microbial species; mostly nonpathogenic) in the human gut whose composition could be manipulated by diet, among other factors [23]. Microbes in the gut are the key to several aspects of human health and disease such as cardiometabolic diseases [24], type 2 diabetes [7], obesity [25], psychiatric disorders [26], non-alcoholic liver disease [27], inflammatory bowel disease [27], and malnutrition [28]. The microbiome of the human gut can play a crucial role in several human functions including producing essential metabolites, digestion, and immune system development [29]. Several edible mushrooms are still an unexploited treasure trove of prebiotic potential and bioactive compounds, as well as positive impacts on the immune system [29].

Overall, the *P. ostreatus* mushroom boasts notable health benefits, including antimicrobial properties and the ability to act as a prebiotic to enhance gut microbiota. This review article highlights the potential roles of the oyster mushroom (*P. ostreatus*) and its antimicrobial and prebiotic activities. The review discusses the role of *P. ostreatus* in modulating the gut microbiota and the importance of gut microbiota for human health. The manuscript will delve into the difference between prebiotics and probiotics.

## 2. Methodology of the Review

A great deal of previous research into oyster mushrooms has focused on the different benefits of such an essential source of nutrients for the human diet. Thus, a large and growing body of literature has investigated the medicinal and pharmaceutical attributes of *P. ostreatus* for human health. Hence, what are the main steps to writing a manuscript (MS) (Figure 1) on the oyster mushroom and its potential for human gut microbiota? After setting the table of contents, we visited different websites and/or the main websites of publishers such as ScienceDirect, Springer, Frontieres, PubMed, Google Scholar, and other international databases. To collect the proper articles for our topic, many keywords were inserted on the search engines, such as “*Pleurotus ostreatus*”, “antimicrobial activity and oyster mushroom”, “prebiotics”, “probiotics”, “prebiotic and *Pleurotus ostreatus*”, “gut microbiota”, “*Pleurotus ostreatus* and gut microbiota”, and so on. The most important factors considered to select suitable articles may include the journal’s reputation, the authors, the publication year, the degree of matching this article with the topic, etc. It was preferable to select the articles within the last decade, and more than 85% of collected articles were published during the last five years. Many tables and figures were included in this MS as a survey and as attractive tools for the readers.

## 3. *Pleurotus ostreatus* and Its Potential

Edible mushrooms are important source of nutrients for human diets due to their enormous health benefits like *Pleurotus ostreatus*. Figure 2 introduces some *P. ostreatus* mushrooms, presented in different forms. *Pleurotus* mushrooms are well-known with several valuable attributes including use in green biotechnology [1], producing bio-nanoparticles [5], medical activities [4], and nutritional and human health [1]. As described in Figure 3, the distinguished properties of *P. ostreatus* are mainly in the nutritional and medicinal activities, such as anticarcinogenic, antioxidative, antiviral, anti-hypercholesteremic, anti-inflammatory, and immune-stimulating properties, which have been attributed to it as a member of the *Pleurotus* genus [30,31]. *P. ostreatus* also has a great ability to bio-degrade organic pollutants by mycoremediation [4,32] or removing sulfonamides from real wastewater [33]. This kind of mushroom can also produce ligninolytic enzymes, which biodelignificate the agro-wastes like cotton stalks [34,35]. Polysaccharides, protein glycoproteins, lectins, high molecular weight-fed fats, small molecule secondary alkaloids, metabolite esters, flavonoids, polyphenols, and triglyceridesols’ bioactive compounds can be found in large numbers in the *P. ostreatus* mushroom [36].

### 3.1. Oyster Mushroom and Agro-Waste Management

The agricultural and food industries are rapidly growing and producing a significant number of byproducts, which have a huge amount of lignocellulose and nutrients. There is an environmental impact related to the production of these byproducts. If not properly handled, these usually end up in municipal landfills contributing to the emission of greenhouse gases and microbiological contamination. Thermal treatment of these byproducts is also used for energy recovery; however, this technology is not efficient enough due to the low calorific value of the organic material. In other worse scenarios, these byproducts that are not properly discarded end up contaminating the water bodies like rivers and lakes. Overall, the environmental impact can be noticed in water, soil, and air quality [34,37,38,39], and the approach of circular economy management becomes relevant when the data show that, for example, fruit byproducts—such as peels, bagasse, seeds, and stems—represent almost 50% of fresh fruit and have high nutritional value, which is sometimes present even in a higher concentration compared to the final product available, and the cost efficiency of these byproducts is different in different years and countries [38]. By 2017, the circular economy approach differed from these traditional practices in the way that the recovery and valorization of wastes and byproducts became a main objective, with the final purpose to put them back in the supply chain to generate economic growth [40].

This mushroom holds significant importance in agricultural waste management [41], as it is a saprophytic mushroom, which can release enzymes and break down dead plant and animal tissues [42]. Agriculture and food byproducts can be valorized for mushroom cultivation, and the spent substrate was shown to be suitable animal feed [43,44]. The usage of these byproducts in the production of *P. ostreatus* urges the implementation of mitigation and compensation strategies for the environment. Moreover, the relative facility to produce this mushroom can help small and medium producers to increase their income since they do not need complex specialized machinery and they can use the residues produced by their agricultural activities [43]. Furthermore, in laboratories, the use of these materials can also be beneficial for research purposes, since they are a cheap alternative to traditional media, which are usually expensive [45]. Since *P. ostreatus* can grow on different lignocellulose substrates, different byproducts from the agricultural and food industry can be revalorized, which allows for using the circular economy approach, where different activities become interconnected and the byproduct or waste of one activity can serve as a raw material for another one [46].

There are many possible agricultural and food industrial waste alternatives, which are shown to be suitable substrates for mushroom fruiting; however, their usage can be a challenge, and proper waste management should be also applied. Several researchers proved the suitability of different agricultural waste products and their combinations for oyster mushroom cultivation; for example, maize stalk and wheat bran [47]; the combination of dried maize stalk, cobs, and wheat straw [48]; a mixture of the brewer’s spent grain and wheat bran [44]; a mixture of fruit peels (avocado peel, banana peel, mango peel, orange peel, pineapple peel, watermelon peel) with wheat straw [49,50]; the combination of maize stalks, bean straw, and maize cob [48,51,52]; and a basal substrate supplemented with *Codonopsis pilosula* stems and leaves have all been tested in previous research [53]. Certain studies indicate that oyster mushrooms cultivated on substrates, such as palm oil bunch, palm oil shaft enriched with wheat bran, and rice bran, exhibit favorable nutritional characteristics, with significant variations observed depending on the specific substrate combinations used [54]. Apart from fruit residues, like apple pomace [55,56], coconut shell [57,58], and grape pomace [59], several other plant wastes like the coffee ground [60], cotton waste and sawdust [51], Faba bean (*Vicia faba* L.) hulls [61], olive pruning residues [62], sugarcane bagasse [63], and wastepaper [64,65] have also been produced and tested and allow for the mushroom mycelia to colonize.

### 3.2. Medicinal and Pharmacological Attributes of P. ostreatus

Medicinal and pharmacological attributes of *P. ostreatus* were confirmed due to its content of bioactive compounds. These bioactive compounds include lovastatin for its anti-hypercholesterolemic activity [66]; β-D glucan, proteoglycans, glycopeptides, and ergothioneine for antitumor properties [7]; lectin, β-D glucan, and ergothioneine for antioxidants [7]; ergothioneine for antidiabetic properties [66]; β-D glucan for antibacterial properties [3]; and ergothioneine for antiviral [2] and anticancer activities [7]. Recently, the prebiotic activity of *P. ostreatus* has become an important tool in the food industry due to its important role to improve and balance the composition of microbiota in the gut, which is linked to the health conditions of the host [67]. Many studies have also been published on the enhancing *P. ostreatus* prebiotic properties for many foods or crops such as olive byproducts [68], riceberry rice [67], and processed cheese [69]. There are many common carbohydrates that are well-known as prebiotics (i.e., inulin, lactulose, fructo-oligosaccharides, galacto-oligosaccharides, and β-glucan) in the human diet [67,70].

The COVID-19 pandemic prompted numerous researchers to focus their efforts on the development of antimicrobial agents sourced from a variety of origins. Bioactive compounds derived from mushrooms are increasingly recognized for their potential to combat emerging infectious diseases and infections caused by antimicrobial-resistant pathogens [14]. Researchers were attracted by mushroom derivatives with naturally occurring antifungal, antibacterial, and antiviral properties. They are suggested as alternative natural preservatives [71,72] and as part of an approach to minimizing antibiotic consumption [73]. The interest in the field of the application of natural antimicrobials as replacements for synthetic additives is growing rapidly, but due to the lack of evidence on their efficiency, more studies are needed in this field [71], especially regarding the incorporation of mushroom derivatives into foods and its impact on the shelf life and safety of the final product [74].

### 3.3. Antimicrobial Activity of Oyster Mushroom

*P. ostreatus* mushrooms are characterized with magnificent medicinal values. The richness and importance of antimicrobial agents (Table 1) in mushrooms have attracted many researchers. Based on the therapeutic properties of the presence of active ingredients, this mushroom can be used as a food supplement to support host health [75]. It worth mentioning that the efficiency of phenolic compounds, like phenolic acids and flavonoids [76], is specific for different species [77] as are their antimicrobial properties [36]. Pleuran (insoluble β-1,3/1,6-D-glucan) is one of the most commonly identified polysaccharides in the *P. ostreatus* mushroom [78,79]. The pharmacological effects (antifungal, antibacterial, and antiviral) of polysaccharides can be determined by the chemical structure and biological effect [75]. Different names have been given to the β-glucan polysaccharides found in different mushroom species (e.g., pleuran, lentinan), which show significant differences in the extent of their bioactive effects [80]; furthermore, this compound has a probiotic regulation effect [81]. The advantage of the application of polysaccharides found in edible mushrooms as antimicrobial agents is that they are easily accessible and free of side effects [67].

According to some recent literature, chitin can be extracted from the cell wall of fungi and should be modified with deacetylation reactions into chitosan, which has better solubility and biocompatibility [18,82,83]; after that, it could be used as an antimicrobial agent with a wide spectrum against Gram-negative and Gram-positive bacteria [18,82] as well as human pathogen fungi [84]. It is worth noting that, while it was first explored from mushrooms, it is mostly found commercially in chitosan produced from crab shells [85].

**Table 1 foods-12-02010-t001:** The medicinal value of *P. ostreatus* mushroom with focus on antimicrobial activity.

Pharmacological Effect	Isolated Bioactive Compounds	Refs.
Antifungal	P-anisaldehyde, chitin, chitosan, 7000 Da of pleurostrin	[8,86]
Antibacterial	Phenolic compounds, tannins, flavonoids, terpenes, β-D Glucan (pleuran), p-anisaldehyde, chitin, chitosan	[2,9]
Antiviral (adjuvant in HBV vaccine)	Lectin (POL)	[10]
Antiviral (HCV)	Laccase	[2,10,14]
Antiviral (HSV-1)	Polysaccharide fraction, β-D-glucan (pleuran), insoluble β-1,3/1,6-D-glucans	[11,13,79]
Antiviral (HSV-2)	Polysaccharide fraction, β-glucan	[14]
Antiviral (COVID-19)	Terpenoids, lectins, glycoproteins, lentinan, galactomannan, and polysaccharides	[14]
Antiviral (HIV)	Ubiquitin-like protein, ergothioneine	[2,14]
Anti-helmintic (gastrointestinal parasites, larvae stage)	Secondary metabolites (alkaloids, flavonoids, phenolic compounds, quinones, peptides, terpenoids, fatty acids)	[15,16]

**Abbreviations:** Hepatitis B virus (HBV), hepatitis C virus (HCV), polygonatum odoratum lectin (POL), herpes simplex virus (HSV-1 and HSV-2), coronavirus disease (COVID-19), human immunodeficiency virus (HIV).

*Pleurotus ostreatus* mushroom fruiting bodies were shown to possess significant antibacterial activities against a wide range of microorganisms (Table 2). These activities are considered an effective against Gram-positive bacteria—such as *Bacillus cereus* [87], *Bacillus pumilis* [9], *Bacillus subtilis* [8], *Enterococcus faecalis* [88], *Micrococcus luteus* [3], and *Staphylococcus aureus* [88]—and Gram-negative bacteria—such as *Burkholderia pseudomallei*, *Enterobacter aerogenes*, *Klebsiella oxytoca*, *Moraxella* sp., *Pseudomonas aeruginosa*, *Salmonella pullorum*) [20], *Escherichia coli* [89], *Klebsiella pneumonia* [88], *Salmonella typhi*, *Shigella* sp. [9], and *Vibrio* sp. [21]—with lower toxicity than antibiotics, but the effectiveness commonly depends on the solvent [2].

*P. ostreatus* mushroom and its derivatives have inhibitory activity on mycelial growth in pathogen fungi, as reported by many researchers (Table 3). Methanol extract of *P. ostreatus* mushroom proved to be effective against Candida strains, like *Candida albicans, Candida glabrata, Candida krusei, Candida parapsilosis, Candida tropicalis* [9], *Epidermophyton floccosum, Pseudomonas aeruginosa* [8], *Microsporum gypseum*, and *Trichophyton rubrum* [91] fungi species. Fewer studies are available regarding the investigation of water extract from *P. ostreatus* mushroom. According to the results, this water extract is efficient against four species: *Candida albicans*, *Cryptococcus humicola*, *Trichosporon cutaneum* [90], and *Penicillium* strain [22] microorganisms. Waktola and Temesgen [8] investigated the effect of hexane-dichloromethane extract of oyster mushrooms, which possesses inhibitor activity against the mycelial growth of *Aspergillus niger* and *Fusarium oxysporum* fungi. The extraction of bioactive compounds from the oyster mushroom with ethyl-acetate has been performed, and its inhibition effect has been demonstrated against *Candida albicans* [20]. The antifungal activity of *P. ostreatus* mushroom extracted with different solvents is commonly linked to the p-anisaldehyde, chitin (chitosan), and pleurostrin bioactive compounds [8,86].

Many studies have reported the antiviral activity of pure extract and purified bioactive compounds from *P. ostreatus* mushroom or mycelia as shown in (Table 4). Water, methanol extract, and β-glucan extracted from *P. ostreatus* showed effective antiviral activities against herpes simplex virus 1 (HSV-1) [93]; however, the sodium-chloride extract was more effective against herpes simplex virus 2 (HSV-2) [79,92]. The pure extract of bioactive compounds (laccase, ubiquitin-like protein) from the *P. ostreatus* fruiting body plays a role in the inhibition of Hepatitis C virus (HCV) and human immunodeficiency virus-1 (HIV-1) [14]. Oyster mushrooms and their aqueous extract may be a role in the prevention of infectious severe acute respiratory syndrome coronavirus 2 (SARS-CoV-2/SARS) diseases and are a source of medicine [12]. Research activities are done to discover new scientific information (Table 4) regarding the antihelmintic activity of the hydroalcoholic and aqueous extract from *P. ostreatus* mushroom fruiting body, mycelia, and harvested substrate. According to some results, oyster mushrooms are effective against gastrointestinal parasites and nematodes [93,94], and this impact is linked to the secondary metabolite compounds, like alkaloids, flavonoids, phenolic compounds, quinones, peptides, terpenoids, and fatty acids [15,16].

## 4. Prebiotics and Their Mechanism

Prebiotics are considered nondigestible food ingredients, which can be consumed by the microbes of probiotics (e.g., live microorganisms including bacteria and yeasts in human gut). Both probiotics and prebiotics have positive roles in promoting human health and his nutrition. Many recent studies have been published on both probiotics and prebiotics with a focus on their mechanisms [97,98,99,100]. Prebiotics also have many direct health benefits, which include oral anti-hyperglycaemic properties and their ability to improve gastrointestinal functions with compounds such as fructooligosaccharides, β-glucan, galacto-oligosaccharides, trans-galacto-oligosaccharides, and inulin [95]. Concerning the prebiotic foods, they include natural and synthetic dietary food products such as asparagus, banana, barley, chicory, garlic, honey, onion, Jerusalem artichoke, soybean, sugar beet, wheat, tomato, cow milk, peas, beans, seaweeds, microalgae, and mushrooms [101]. Human consumption of prebiotics of diets can improve food intake control and reduce human body fat content and weight gain [102] (Figure 4).

Many specific prebiotics can promote the growth of gut microbiota by protecting and/or promoting producing beneficial fermentation products. Furthermore, these prebiotics have not only the ability to protect the gastrointestinal system, but also the other organs of the body, including the cardiovascular, central nervous, and immune systems [99]. The interaction among prebiotics, probiotics, and the gut microbiota is a complex process, which has boosting effects on gut immunity and the mechanisms of interaction [97]. These mechanisms can be summarized in the following model of prebiotic benefits to human health:Reducing risk of infection by resisting pathogens and increasing natural killer activity;Improving mineral absorbability by lowering pH and enhancing soubility of minerals;Improving bowel function by increasing water binding;Modulation of immune cells and reducing allergies;Improving intestinal barrier function and lowering blood lipid levels;Reducing inflammation and maintaining tight junction integrity; andIncreasing satiety by increasing leptin and lowering ghrelin, two hormones that control the human appetite and fullness [97,99].

## 5. Prebiotic Activities of *Pleurotus ostreatus*

Mushrooms are rich sources of bioactive polysaccharides and many essential amino acids, as well as a potential candidate for prebiotics such as chitin, galactans, glucans, mannans, krestin, lentinan, hemicellulose, and xylans [104]. In addition to dietary fiber, various proteins play an important role in modulating the gut microbiome [105], along with phenolics [106,107]. The most essential chemical constituent present in this mushroom is β-glucan [(1,3)/(1,6)-glucans], which is responsible for its antimicrobial [34] and prebiotic functions [25,108]. *P. ostreatus* mushroom is rich in β-glucan, which can be linked to proteins. The other significant compound is chitin along with its derivatives, which have been shown to modulate the gut microflora in such a way that the compound gives energy and carbon source for the probiotics (digestion, metabolization) [17,18,82]. The efficiency of phenolic compounds is specific for different species [78], and their prebiotic functions are well-known in the food and pharmaceutical industry as well [25,108]. The importance of the utilization of the prebiotic properties of edible mushrooms is growing rapidly in the food industry. They are used because of their prebiotic properties (oligosaccharides or polysaccharides) and to produce symbiotics (combination of probiotics and prebiotics) [109].

Numerous types of research have demonstrated the extraction of prebiotic substances from *P. ostreatus* mushroom fruiting body, substrate, and mycelia, which proved to be very useful for probiotic growth stimulation and microbiota manipulation. *P. ostreatus* mushroom fruiting body was shown to have prebiotic activity (Table 5) by stimulating the growth of probiotic bacteria strains (*Lactobacillus acidophilus*, *Lactobacillus plantarum*) stronger than commercial probiotics like inulin and fructo-oligosaccharide [110]. A few studies on the human gut microbiome revealed that the *P. ostreatus* mushroom increased the *Bacterodies* spp. [111], *Bifidobacterium* spp. [112,113], and *Faecalibacterium prausnitzii* microorganism [112]. Polysaccharides extracted from *P. ostreatus* fruiting body modulated the growth of *Bifidobacterium* spp., *Enterococcus* spp. [114], and *Lactobacillus* spp. [113,115]. The mycelia also have prebiotic properties. According to the results of a fermentation experiment, the prebiotic properties of germinated riceberry rice (*Oryza sativa* L.) can be strengthened by the addition of mycelia. The results showed the growth promotion of *Pediococcus* spp. and *Streptococcus lactis* [67].

## 6. Gut Microbiota and Its Importance

The human gut is a very important organ that possesses millions of microorganisms as a complex microbial community and forming its multi-directional connecting axes with other organs [23]. Different information on human gut microbiota are presented in Figure 5, whereas Figure 6 includes a list of some metabolites that are produced by the gut microbiota and their functions. Gut microbiota as an expression refers to the community of different species of microorganisms in human gut, whereas gut microbiome expresses the different collective genomes of the microorganisms in the human gut [103]. The manipulation of the gut microbiota with prebiotics could be an excellent strategy to prevent diseases and disorders and maintain normal intestinal homeostasis (Figure 7). The most common phyla in the gut microbiota are Firmicutes and Bacteroidetes [111,117]. The ratio between them plays a significant role in the risk of obesity, aggravating intestinal dysfunction, and systemic injury [118]. The Firmicutes phylum has genetic characteristics that contribute to the fermentation of dietary fibers and thus to the health of the host [119]. The communication between the gut microbiome and the brain (nervous system) is a very close and strict network [120,121] that plays a significant role in several neurological disorders [108].

The importance of nondigestible dietary fiber intake is growing rapidly, due to its extending effect on the gut microbiome and health, especially intestinal health, immune modulation, pathogen invasion, and metabolic side effects [122,123]. If a substance meets the criteria for being a prebiotic, the consequences for gut microbiota and human health are shown in Figure 8. Prebiotics should be resistant to the upper gastrointestinal tract, reaching the colon, fermentation by probiotics, and selective stimulation of the growth of the intestinal bacteria [124]. During the fermentation of probiotics, the pH value is decreased, which creates an environment that is disadvantageous for the growth of pathogens and has shown to be supportive for beneficial bacteria. The good bacteria produce B1, B2, B3, B6, B8, B9, and B12 vitamins. As a consequence, the vitamin and mineral absorption and the forming of organic acids and amino acids will be enhanced. Short-chain fatty acids have effects on gastrointestinal epithelial cell integrity, immunity, glucose homeostasis, lipid metabolism, body weight control, and the reduced risk of some types of cancer [122].

The antibiotic resistance growing number of patients with bacterial infections is placing substantial health and economic burden worldwide [125]. Connections were found between antibiotic consumption and resistant strains in hospitals [126]. The oral administration of prebiotics has effects on the selective promotion of the proliferation and activity of good bacteria in the colon [126]. Mushroom poly- and oligosaccharides are able to alter the immune system; prevent colonic cancer; and reduce pathogen invasion, the risk of intestinal disease, cardiovascular disease, non-insulin-dependent diabetes, and osteoporosis obesity [70], thus enhancing the mineral density at the bone [127].


Figure 8Nondigestible dietary fiber intake is often mentioned for its role in extending the effects on gut microbiome and human health, especially when it completes the criteria for being a prebiotic. **Abbreviation:** short-chain fatty acids (SCFAs) **Sources: [23,103,108,111,117,119,120,121,122,123,124,128]**.
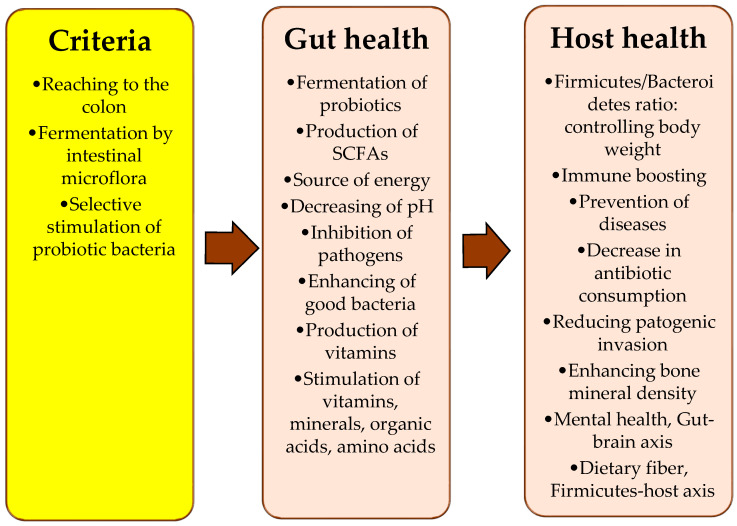



## 7. *Pleurotus ostreatus* and Gut Microbiota

The relation between mushroom and human gut microbiota is a crucial for human health and diseases [129]. Mushrooms are rich in many bioactives and dietary fiber, which can promote human health by producing many vital compounds like short-chain fatty acids [130]. The composition of gut microbiota can be preferably changed by the consumption of mushroom polysaccharides [108]. Mushroom polysaccharides (long chains of monosaccharides linked by glycosidic bonds and oligosaccharides (short chains of carbohydrates of 3 to 10 monomers), like chitin [131], β-glucan, inulin [132], α-glucan, and galactomannans [133], as well prebiotics, increase the production of short-chain fatty acids and favor that of some beneficial genera [134,135], for example *Lactobacilli* and *Bifidobacteria* [136]. The most common short-chain fatty acids (SCFAs) are acetate, propionate, butyrate [122], and lactate [121].

Dietary fibers are complex carbohydrate polymers with three or more monomeric units that are resistant and promote the production of short-chain fatty acids (SCFA), improving the intestinal mucosal barrier, regulating lipid metabolism, activating specific signaling pathways [137], and modifying the composition of colonic microbiota, which confers various health benefits to the host [138]. Correlations have been found between biological activity and the molecular ratio of mushroom polysaccharides, the composition of monosaccharides, and the type of glycosidic bonds [137]. The water-soluble dietary fibers play an important role at the protection against obesity and diseases [138]. Edible mushrooms are a valuable source of soluble and insoluble dietary fiber [139]. The most important dietary fibers from the mushroom may include (1→3)-β-D-glucan, chitin, mannan [46], xylan, and galactan [140]. Polysaccharides promote intestinal microbial biodiversity, balance the composition of the microbiota, promote the growth of probiotic bacteria and fungi, and improve immune status [48]. The dietary fiber content of edible mushrooms varies according to developmental level and species, and they show a different distribution in different fractions [141].

The human gastrointestinal tract has a diverse array of microorganisms, which play potential roles in human health and diseases. Many recent studies have been published on the impacts of the oyster mushroom and *Pleurotus* spp. on gut microbiota from different points of view as follows:The golden oyster mushroom (*Pleurotus citrinopileatus*) has distinguished pharmacological functions like modulation of hepatoprotective and human gut microbiota through the protective effects of polysaccharide peptides. These bioactives of polysaccharide peptides can extracted from this mushroom and metabolized by the gut microbiota to produce short-chain fatty acids, which promote the functions of liver [142];The impacts of polysaccharides (β-1,6-glucan) extracted from the fruiting bodies of *Pleurotus eryngii* were investigated on regulating immunity and promoting the gut microbiota [143,144];A study was conducted on the influence of isolated glucopyranose from mycelium of *Pleurotus geesteranus* on preventing alcoholic liver diseases and the gut microbiota. The mechanism of this bioactive compound may cause a balance in the gut–liver axis by increasing intestinal tight junction proteins, thus elevating the abundance of short-chain fatty acids producers in the intestine by regulating the composition of gut microbiota [145];The role of *Pleurotus ostreatus* in ameliorating obesity in obese mice and modulating the gut microbiota was reported. This mushroom can induce the gut microbiota functions by upregulating the metabolism of lipids and carbohydrates and the biosynthesis of bile acids, as well as downregulating the signaling pathway of adipocytokine and the biosynthesis of steroid hormones [108];The role of oyster mushroom (*Pleurotus sajor-caju*) in modulating gut microbiota in Zucker rats as a prebiotic agent was studied by enhancing the growth of SCFA-producing bacterial genera (e.g., *Blautia*, *Bifidobacterium*, *Faecalibaculum*, and *Roseburia*), while decreasing the abundance of *Escherichia–Shigella* [146]; andThe role of *Pleurotus eryngii* was studied as a prebiotic agent in promoting the gut microbiota via in vitro fermentation in presence of selenium, which increases lead adsorption by bacteria of *Desulfovibrio*, leading to a reduction of lead toxicity in humans [147].

## 8. General Discussion

A general discussion on the main topics can be presented in this section through answering the following questions: What is the importance of the *P. ostreatus* mushroom?

What are the most important prebiotic and antimicrobial compounds in the *P. ostreatus* mushroom? What are the main differences between prebiotics and probiotics? What is the relationship between oyster mushrooms and prebiotics? What is role of oyster mushrooms on the modulation of gut microbiota?

With respect to the first research question on the importance of the *P. ostreatus* mushroom, this mushroom is commonly known as the oyster with various benefits in medicinal treatments [71,74], agro-waste management [38,46], and human digestion [1]. *P. ostreatus* is one of the few mushroom species that can grow on several lignocellulosic agro-wastes [38]. *P. ostreatus* is also an important food source, and it is highly nutritious; rich in protein, dietary fiber, vitamins, and minerals; and low in calories and fat content, making it a healthy addition to the diet [31]. Its versatile properties make it a promising candidate for developing functional foods, nutraceuticals, and medicines, while its ability to grow on agro-waste and provide a sustainable solution to waste management makes it a valuable resource for sustainable development [38,71].

On the other hand, the *P. ostreatus* mushroom is rich in antimicrobial agents, including phenolic compounds, flavonoids, and polysaccharides, which can be used as food supplements for human health benefits. Polysaccharides found in edible mushrooms are a safe and easily accessible source of antimicrobial agents with no side effects [66]. Chitin, extracted from the cell wall of fungi, and modified into chitosan, is another antimicrobial agent that can be used against Gram-negative and Gram-positive bacteria and human pathogenic fungi. It worth knowing that the *P. ostreatus* mushroom has significant antibacterial, antifungal, and antiviral activities with lower toxicity than antibiotics. Bioactive compounds, such as p-anisaldehyde, chitin, and pleurostrin, have been identified as contributing to these activities. Studies have demonstrated the extraction of prebiotic substances (chitin and glucan) from different parts of the *P. ostreatus* mushroom, which have been useful for stimulating the growth of probiotic bacteria strains and manipulating the microbiota, and support host health [18,82,86]. The utilization of the prebiotic properties of edible mushrooms is growing rapidly in the food industry. Further research is needed to understand the effect of different mushroom derivatives against living organisms and develop new and cost-efficient procedures for the extraction of active compounds [79,82,115].

With respect to the third research question on the probiotics, they include the different species of microbes living in the gastrointestinal system (mainly beneficial and natural gut microbiota) and are supported by the prebiotic sources to grow and develop. These probiotics could be broadly classified into groups of *Lactobacilli*, *Bifidobacteria*, and others [99]. The main positive functions of probiotics on the human body may involve regulating human intestinal health and immune function, improving the metabolism of blood lipid and its sugar, maintaining the balance of microbiota, and helping human body to better digest and absorb food residues [99].

Furthermore, prebiotics are important compounds that support and promote the activity of probiotics in the human gut including oligosaccharide carbohydrates, polysaccharides, polyphenols, and polypeptide polymers. For example, polyphenols extracted from blueberries can reduce weight and normalize lipid metabolism [148], whereas extracted polysaccharides from algae can improve the activity of and produce functional metabolites in the intestinal microbiota [149].

Concerning the relationship between oyster mushrooms and prebiotics, an increased concern was noticed in the relationship between prebiotics (e.g., β-glucan and inulin) extracted from oyster mushrooms and their potential probiotics [132]. This synergistic relationship could be applied in several applications such as the formulation of synbiotic microcapsules [132], as bioresources for superfoods [150], for disaster relief situations [151], for developing an industry of edible mushroom protein as meat analogues [152], for producing innovative mushroom products [153], and for processed cheese that contains high antioxidant activity [69]. The ecological perspective of probiotics and prebiotics is needed to investigate supporting the microbiome by using certain probiotic strains, which can reduce the absorbed amounts of heavy metals from foods in polluted regions worldwide [98]. This approach can be helpful also for removing toxins and drugs from our food and environment as well.

What about the role of oyster mushrooms on modulation of gut microbiota? It is well-documented that the gastrointestinal tract in human has a lot of microbes that are crucial in controlling human health, and diseases caused by their imbalance in the gut microbiota, namely dysbiosis, causing several other diseases. The human consumption of oyster mushrooms can supply the gut with essentials prebiotics for the activity of probiotics over there, whereas the imbalance in the gut microbiota leads to many human diseases, such as cancer [129] and hepatic diseases [142]. Mushroom foods can positively control the human gut microbiota due to their distinguished content of dietary fiber, which is considered beneficial for its antiobesity, antitumor, and immunomodulatory properties [130]. There are many suggested mechanisms, which may involve changes in the human gut microbiota, increases in the level of short-chain fatty acids, promoting the production of immunoglobulin A, and promoting the host immune system [130]. Therefore, oyster mushrooms are considered healthy foods for future generations.

As a result, the future of probiotics [154] and prebiotics should include emphasizing the need for standardized testing methods and personalized approaches to optimize their effectiveness [155,156,157]. There is a relationship between prebiotics, the gut microbiota, and metabolic risks, suggesting that prebiotic consumption could potentially improve metabolic health through modulation of the gut microbiota [158,159,160,161]. Many researchers have investigated the improvement of the highly complex gut microbiota ecosystem, which can boost immunity and help to prevent against several diseases when it is supported, and researchers are also working on the development of new methods for the prevention of several diseases [162,163,164,165].

## 9. Conclusions

The antimicrobial compounds of oyster mushroom (*Pleurotus ostreatus* (Jacq. ex Fr.)) are shown to have wide spectrum application against bacteria, fungi, viruses, and gastrointestinal parasites. Furthermore, the pharmacological usage of these compounds is a successful tool to reduce pathogenic invasion and improve the gut microflora. Most studies focused on β-glucan (pleuran) and how it can be extracted from the fruiting body, mycelia, and substratum. Chitin is also an important prebiotic compound; however, it should be stabilized and formed into chitosan to reach better efficiency. In the future, it will be interesting to explore the potential applications of new cost-efficient methods for extracting and analyzing prebiotic and antimicrobial compounds from oyster mushrooms and other mushroom derivatives, like spent mushroom substrate. Moreover, investigating the effects against living organisms will be important in advancing our understanding of these compounds and their potential therapeutic applications.

## Figures and Tables

**Figure 1 foods-12-02010-f001:**
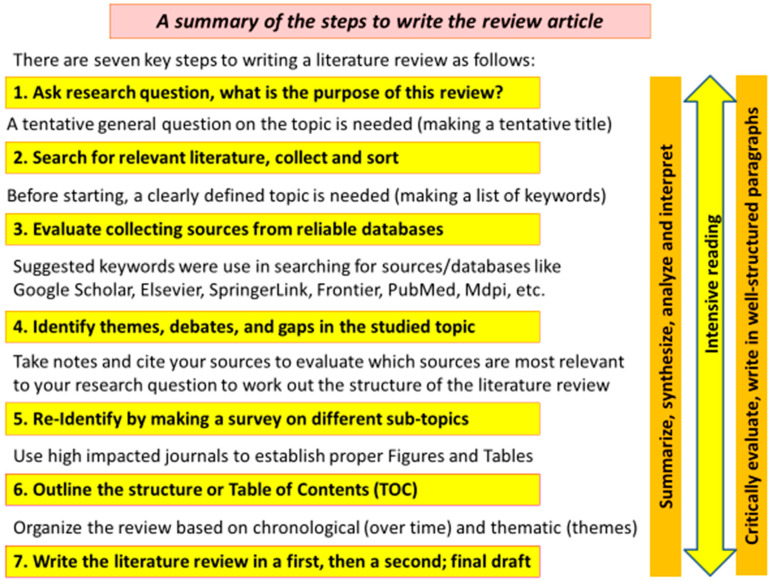
An overview of the key stages involved in writing a review article.

**Figure 2 foods-12-02010-f002:**
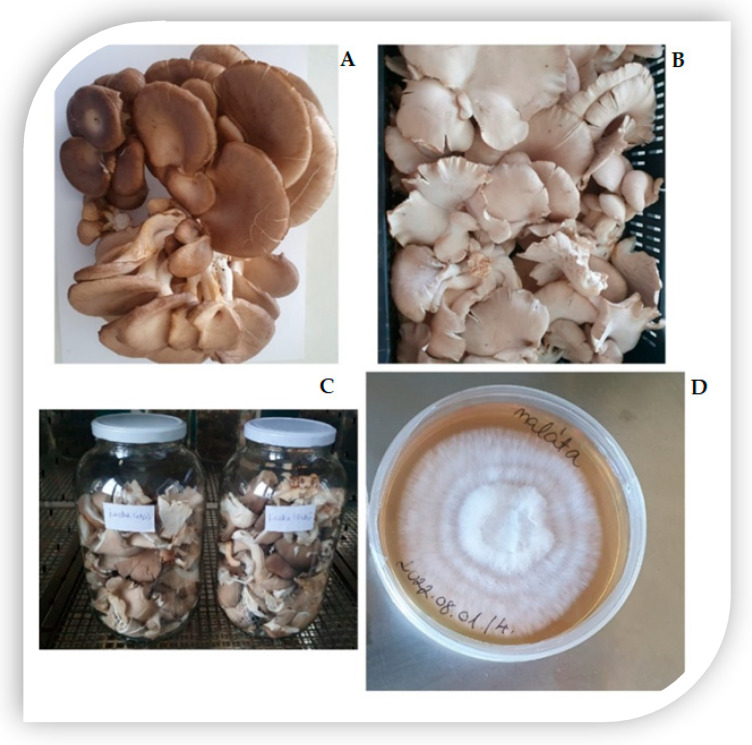
An overview is given on some *P. ostreatus* mushrooms, including: (**A**) fresh samples that were obtained from a household farming in Tiszavasvári, (**B**) fresh samples of *P. ostreatus* mushrooms that were obtained from Magyar Gomba Kertész Kft. in Demjén, (**C**) sliced mushroom in glass jars that comes from Magyar Gomba Kettész Kft. in Demjén, and (**D**) *P. ostreatus* mycelia that were grown on malt extract media from Magyar Gomba Kertész Kft. in Demjén. All photos were taken by Gréta Törős in Nano-Food Lab, Debrecen.

**Figure 3 foods-12-02010-f003:**
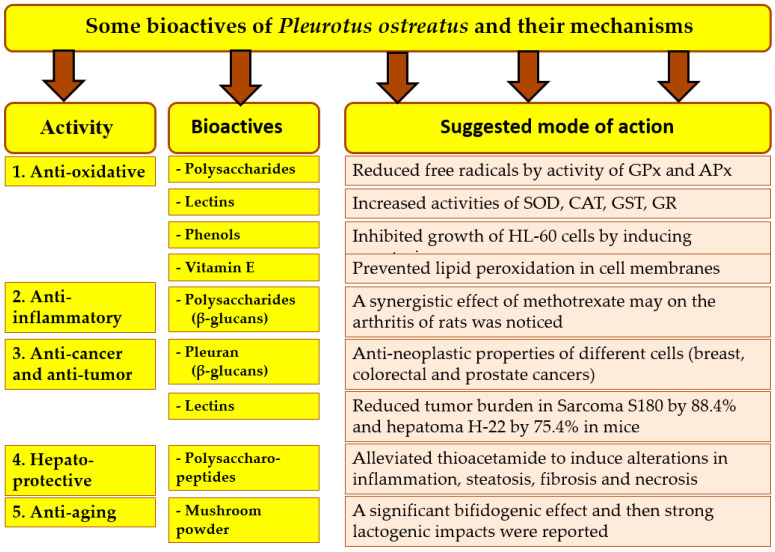
A list of some bioactives of *Pleurotus ostreatus* and their mechanisms. **Abbreviations:** Glutathione peroxidase (GPx), superoxide dismutase (SOD), catalase (CAT), ascorbate peroxidase (APx), glutathione reductase (GR), and glutathione S-transferases (GST). **Sources: [1,30,31]**.

**Figure 4 foods-12-02010-f004:**
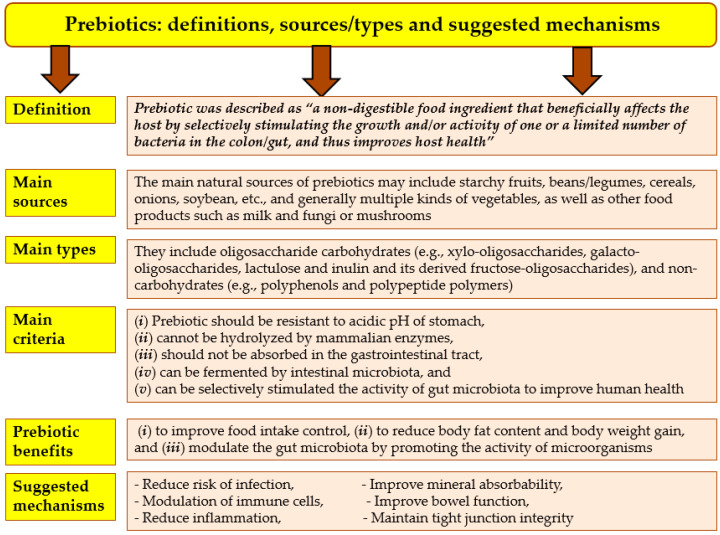
The definition of prebiotic, their sources, types, and possible mechanisms. **Sources:** [97,99,101,102,103].

**Figure 5 foods-12-02010-f005:**
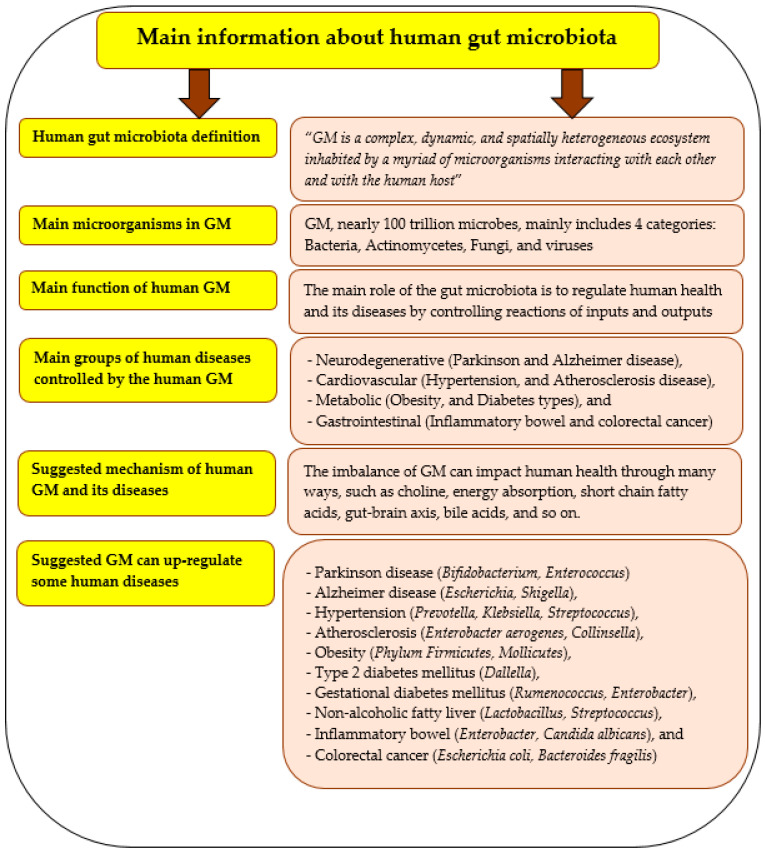
Some basic information on human gut microbiota including the definition, function, mechanism of function, and their control of human diseases. **Sources**: [27,103].

**Figure 6 foods-12-02010-f006:**
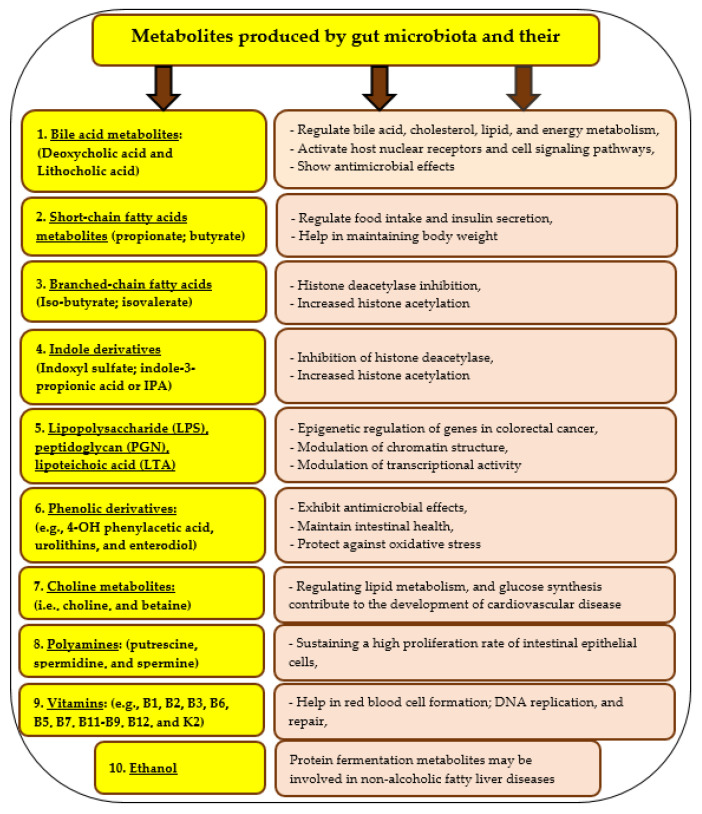
A list of some metabolites that produced by gut microbiota and their functions. **Source:** [23].

**Figure 7 foods-12-02010-f007:**
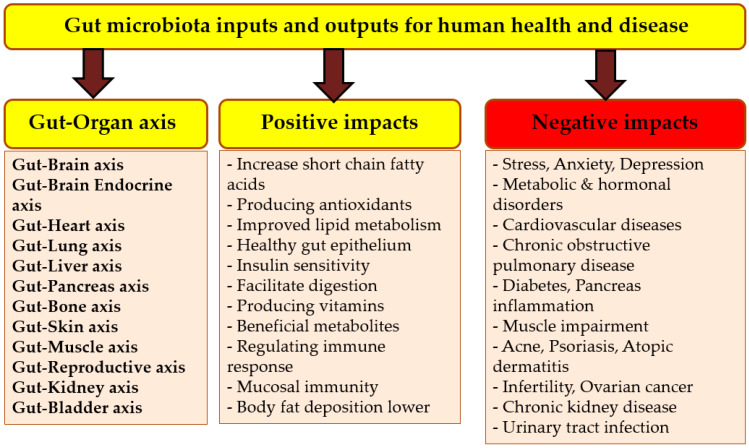
The main outputs and inputs including the positive and negative impacts of the gut microbiota on human health. **Sources: [23,27,103]**.

**Table 2 foods-12-02010-t002:** Antibacterial activity of *P. ostreatus* mushroom and mycelia extracted with different solvents.

Used Solvent(s)	Anti-Microbes (Bacteria)	Refs.
Chitosan	*Bacillus cereus*	[87]
Methanol	*Bacillus pumilus*	[9]
Hexane-dichloromethane	*Bacillus subtilis*	[8]
Ethyl acetate	*Burkhoderia pseudomallei*, *Enterobacter aerogenes*	[20]
Methanol	*Enterococcus faecalis*	[88]
Ethyl acetate, methanol, water extract, ethanol: chloroform: distilled water (2:2:3), plus (75%) of chloroform, ethanol, and acetone	*Escherichia coli*	[3,21,89,90]
Ethyl acetate	*Klebsiella oxytoca*	[20]
Methanol	*Klebsiella pneumonia*	[88]
Ethanol, chloroform, and distilled water, in a constant ratio of 2:2:4	*Micrococcus luteus*	[3]
Ethyl acetate	*Moraxella* sp.	[20]
Ethyl acetate, methanol, and Hexane-dichloromethane	*Pseudomonas aeruginosa*	[8,20]
Ethyl acetate	*Salmonella pullorum*	[20]
Methanol	*Salmonella typhi*	[9]
Water and methanol	*Shigella* sp.	[9,21]
Ethyl acetate, methanol, ethanol: chloroform: distilled water (2:2:1), water plus (75%) of chloroform, ethanol, and acetone	*Staphylococcus aureus*	[3,21,89,90]
Water	*Vibrio* sp.	[21]

**Table 3 foods-12-02010-t003:** Antifungal activity of *P. ostreatus* mushroom and mycelia extracted with different solvents.

Used Solvent(s)	Anti-Microbes (Fungi)	Refs.
Hexane-dichloromethane	*Aspergillus niger*	[8]
Ethyl acetate, methanol, ethanol: chloroform: water (2:2:2), and water	*Candida albicans*	[3,9,90]
Methanol	*Candida glabrata*, *C. krusei*, *C. parapsilosis*, *C. tropicalis*	[9]
Water	*Cryptococcus humicola*, *Trichosporon cutaneum*	[92]
Methanol	*Epidermophyton floccosum*, *Microsporum gypseum*, *Trichophyton rubrum*	[91]
Hexane-dichloromethane	*Fusarium oxysporum*	[8]
Water	*Penicillium* sp.	[21]

**Table 4 foods-12-02010-t004:** Antihelmintic and antiviral activity of *P. ostreatus* mushroom and mycelia extracted with different solvents.

Activity	Used Solvent(s)/Bioactives	Microbe/Disease	Refs.
Antihelmintic	Hydroalcoholic extract	Gastrointestinal nematodes	[94]
Antihelmintic	Aqueous extract	*M. incognita* parasitic nematode	[93]
Antiviral	Laccase purified from *P. ostreatus* mushroom fruiting	HCV	[10,14]
Antiviral	Ubiquitin-like protein purified from *P. ostreatus*	HIV	[2,14]
Antiviral	Water extract	HSV-1	[92]
Antiviral	Extracted β-glucan	HSV-1	[79]
Antiviral	Aqueous and methanol extracts of *P. ostreatus*	HSV-1	[95]
Antiviral	Sodium-chloride extract of *P. ostreatus* mycelia	HSV-2	[92]
Antiviral	Aqueous extract of *P. ostreatus* mushroom	SARS-CoV-2/SARS	[11]
Antiviral	Aqueous and methanol extracts of *P. ostreatus*	Human Cytomegalovirus	[96]

**Abbreviations:** Hepatitis C virus (HCV), herpes simplex virus (HSV-1 and HSV-2), severe acute respiratory syndrome coronavirus 2 (SARS-CoV-2), severe acute respiratory syndrome (SARS).

**Table 5 foods-12-02010-t005:** The medicinal value of *P. ostreatus* mushroom with focus on prebiotic activity.

Prebiotic Activity	Probiotic Microbes	Refs.
High content of total carbohydrates and total reducing sugar as indicator for prebiotic yield	*Lactobacillus acidophilus*, *Lactobacillus plantarum*	[116]
β-glucans of *P. ostreatus* enhanced the gut microbiota of 65-year-old humans	*Bifidobacterium* spp., *Faecalibacterium prausnitzii*	[112]
Polysaccharides of *P. ostreatus* mushroom enhanced the gut microbiota in humans	*Bifidobacterium* spp.	[113]
Polysaccharides from *P. ostreatus* mushroom promoted the activity of studied microbes	*Lactobacillus* spp., *Enterococcus* spp.	[113,114]
Polysaccharides from *P. ostreatus* promoted the activity of studied microbes	*Bifidobacterium* spp.	[116]
β-glucans in the *P. ostreatus* mushroom promoted the activity of studied microbes	*Lactobacillus* spp.	[115]
β-glucans in *P. ostreatus* mushroom enhanced the gut microbiota in humans	*Bacteroides* spp.	[111]
γ-aminobutyric acid and β-glucan in *P. ostreatus* improved digestive system health	*Pediococcus* spp., *Streptococcus lactis*	[67]
Polysacharrides extracted from *P. ostreatus* promoted the activity of studied microbes	*Lactobacillus plantarum*, *Lactobacillus acidophilus*	[96]

## Data Availability

Not applicable.

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
