# Peer review of "Modulation of the Gut Microbiota with Prebiotics and Antimicrobial Agents from Pleurotus ostreatus Mushroom"

_foods, 2023, doi:10.3390/foods12102010_

Round 1

Reviewer 1 Report

Dear Editors and authors,

1-This study primarily investigated the antibacterial and prebiotic activity of oyster mushroom, with a focus on prebiotics in the abstract. Prebiotics do not frequently participate in antimicrobial resistance. Therefore, the manuscript abstract needs to be revised to align with the research content and be clear to the reader.

2-Many scientific names are not written in italics. Please review the manuscript and write any scientific name of bacteria or plants in italics.

3-The keywords in the manuscript should reflect the content of the research, and I did not find the keyword "Prebiotic compound" which should be included.

4-Some figures in the manuscript are unclear and cannot be understood, and were not explained, such as Figure 1B. The authors must provide an explanation for each figure to make it clear to the reader.

5-Abbreviating some terms is incorrect. The term should be written first, and then abbreviated, rather than writing the abbreviation or symbol before the term, see line 111.

6-The tables in the manuscript need to be formatted and rearranged. Please refer to Table 2.

7-Some figures in the manuscript are unclear, such as Figure 7. It needs to be redrawn or clarified to make it clear to the reader.

9-The discussion is unclear as it is not a detailed reiteration of the results. It should interpret the results and explain the reasons behind them. Many points were discussed, but in a manner of restating the results. Please rephrase the discussion. See Modulation of the Gut Microbiota with Prebiotics and Antimi-crobial Agents

10-The manuscript needs some scientific references to support it, such as

-        Al-Sahlany, S. T., & Niamah, A. K. (2022). Bacterial viability, antioxidant stability, antimutagenicity and sensory properties of onion types fermentation by using probiotic starter during storage. Nutrition & Food Science52(6), 901-916.

-        Cunningham, M., Azcarate-Peril, M. A., Barnard, A., Benoit, V., Grimaldi, R., Guyonnet, D., ... & Gibson, G. R. (2021). Shaping the future of probiotics and prebiotics. Trends in microbiology29(8), 667-685.

-        Yasmin, Adeela, et al. "Prebiotics, gut microbiota and metabolic risks: Unveiling the relationship." Journal of functional foods 17 (2015): 189-201.

Author Response

Dear Reviewer 1#

Many thanks for your time and efforts to improve our MS to be ready for publication!

1-This study primarily investigated the antibacterial and prebiotic activity of oyster mushrooms, with a focus on prebiotics in the abstract. Prebiotics do not frequently participate in antimicrobial resistance. Therefore, the manuscript abstract needs to be revised to align with the research content and be clear to the reader.

Response: Many thanks for your comments! The abstract has been revised to provide clarity to the reader and to highlight the potential of prebiotics and the antimicrobial agent found in Pleurotus ostreatus mushroom. Furthermore, the relationship between prebiotics and antibiotic resistance is clearly described.

-Many scientific names are not written in italics. Please review the manuscript and write any scientific name of bacteria or plants in italics.

Response: Thanks for your comments! We checked the MS again and Italic names were corrected.

3-The keywords in the manuscript should reflect the content of the research, and I did not find the keyword "Prebiotic compound" which should be included.

Response: Done, Thanks!

4-Some figures in the manuscript are unclear and cannot be understood, and were not explained, such as Figure 1B. The authors must provide an explanation for each figure to make it clear to the reader.

Response: Thanks for your comment! The explanation for Figures 1 and 2 was corrected. The title of Figure 1. is explained for better understanding. The color of the tables and some syntax error changed for the better understanding.

5-Abbreviating some terms is incorrect. The term should be written first, and then abbreviated, rather than writing the abbreviation or symbol before the term, see line 111.

Response: Thanks for your comment! Abbreviations were correctly listed after the full name or term.

6-The tables in the manuscript need to be formatted and rearranged. Please refer to Table 2.

Response: Thanks for your comments! We checked all tables were adjusted, thanks!

7-Some figures in the manuscript are unclear, such as Figure 7. It needs to be redrawn or clarified to make it clear to the reader.

Response: Thanks for your comment! All Figures re-formatted again including Figure 7, which was described and the sequence of the figure has been changed and the title of this figure was re-described for better understanding.

9-The discussion is unclear as it is not a detailed reiteration of the results. It should interpret the results and explain the reasons behind them. Many points were discussed, but in a manner of restating the results. Please rephrase the discussion. See Modulation of the Gut Microbiota with Prebiotics and Antimicrobial Agents.

Response: Thanks for your comment! The general discussion sectioned was revised and extended with 2 more questions regarding with the most important antimicrobial and prebiotic bioactive compounds. The discussion and conclusion were extended and clarified in the revised MS.

10-The manuscript needs some scientific references to support it, such as

-        Al-Sahlany, S. T., & Niamah, A. K. (2022). Bacterial viability, antioxidant stability, antimutagenicity and sensory properties of onion types fermentation by using probiotic starter during storage. Nutrition & Food Science52(6), 901-916.‏

-        Cunningham, M., Azcarate-Peril, M. A., Barnard, A., Benoit, V., Grimaldi, R., Guyonnet, D., ... & Gibson, G. R. (2021). Shaping the future of probiotics and prebiotics. Trends in microbiology29(8), 667-685.‏

-        Yasmin, Adeela, et al. "Prebiotics, gut microbiota and metabolic risks: Unveiling the relationship." Journal of functional foods 17 (2015): 189-201.‏

Response: Thanks for your suggestion! The suggested literatures were added.

Many thanks for your time, hoping our correction will be accepted, thanks!

Reviewer 2 Report

The authors mainly studied the effects of antibacterial and prebiotics activity of oyster mushroom on the intestinal flora. This study is interesting, However, there are some points that need to be clarified. Recommendations were published after the revision.

1. This study mainly investigated the antibacterial activity and prebiotics activity of oyster mushroom, while in the abstract focuses on the prebiotic’s activity, which is very little involved in antimicrobial resistance.

2. Line81-82, Keyword search Should joinprobiotics and Pleurotus ostreatus

3. Line51-53. In fact, there is positive correlation about phenols and probiotics regulation effect (Food Bioscience. 50(2022): 101946.).  

4. Line105. We cannot understand the description of Figure 1 B, so it is suggested to make it clear.

5. Line 111,The abbreviation is incorrect, “GR (glutathione reductase)” alternative to “glutathione reductase (GR)”.

6. Line187. β-Glucan has probiotics regulation effect (Food & Function, 2022, 13(24), 12686-12696.).

7. Table 3 spans the page, and you should mark the Continuation table.

8. Table 2 has four lines, should pay attention to the chart format. Please use three lines.

9. The format of Figure 3 is chaotic, with different font sizes and positions.

10. The text narrative in Line 365-372 should not be between Figure 6 and Figure 7.

11. Fig 7. It is difficult for people to watch, so the description in the figure should be uniformly vertical.

12. Line 429. Lack of content. There should also be content after the word “and”.

13. The discussion is not a redescription of the results and lacks logic. The title of the authors is Modulation of the Gut Microbiota with Prebiotics and Antimi-crobial Agents from Pleurotus Ostreatus Mushroom, while it is not reflected in the discussion.

14. The reference should be updated in recent years.

Author Response

Dear Reviewer 2#

Many thanks for your time and efforts to improve our MS to be ready for publication!

The authors mainly studied the effects of the antibacterial and prebiotic activity of oyster mushrooms on the intestinal flora. This study is interesting, However, there are some points that need to be clarified. Recommendations were published after the revision.

Response: Thanks for your comment! Many thanks for your encouragements as well.

  1. This study mainly investigated the antibacterial activity and prebiotics activity of oyster mushroom, while in the abstract focuses on the prebiotic’s activity, which is very little involved in antimicrobial resistance.

Response: Thanks for your comment! The abstract has been revised to provide clarity to the reader and to highlight the potential of prebiotics and the antimicrobial agent found in Pleurotus ostreatus mushroom. The relationship between prebiotics and antibiotic resistance is clearly described.

  1. Line 81-82, Keyword search should join“probiotics and Pleurotus ostreatus.

Response: Thanks for your comment! The suggested keywords have been added, thanks!

  1. Line 51-53. In fact, there is positive correlation between phenols and probiotics regulation effect (Food Bioscience. 50(2022): 101946.).

Response: Thanks for your comment! Added as ref. no 22 in the revised MS!

Dong, L.; Qin, C.; Li, Y.; Wu, Z.; Liu, L. Oat phenolic compounds regulate metabolic syndrome in high fat diet-fed mice via gut microbiota. Food Biosci, 2022, 50, Part A, 101946. https://doi.org/10.1016/j.fbio.2022.101946.

  1. Line 105. We cannot understand the description of Figure 1 B, so it is suggested to make it clear.

Response: Thanks for your comment! The title of Figure 1. is explained for better understanding.

  1. Line 111,The abbreviation is incorrect, “GR (glutathione reductase)” alternative to “glutathione reductase (GR)”.

Response: Thanks for your comment! Corrected.

  1. Line 187. β-Glucan has probiotics regulation effect (Food & Function, 2022, 13(24), 12686-12696.)

Response: Thanks for your comment! Thanks. Your suggested article was inserted in the revised MS in ref. no. 144!

Li, Y.; Qin, C.; Dong, L.; Zhang, X.; Wu, Z.; Liu, L.; Yang, J.; Liu, L. Whole grain benefit: synergistic effect of oat phenolic compounds and β-glucan on hyperlipidemia via gut microbiota in high-fat-diet mice. Food Funct., 2022,13, 12686-12696.

  1. Table 3 spans the page, and you should mark the “Continuation table”.

Response: Thanks for your comment! The table was corrected and replaced into the next page.

  1. Table 2 has four lines, should pay attention to the chart format. Please use three lines.

Response: Thanks for your comment! Was checked again, Done, thanks!

  1. The format of Figure 3 is chaotic, with different font sizes and positions.

Response: Thanks for your comment! The all figures have been fixed for better understanding.

  1. The text narrative in lines 365-372 should not be between Figure 6 and Figure 7.

Response: Thanks for your comment! The text was replaced above the Figure 6 (corrected into Figure 7.).

  1. Fig 7. It is difficult for people to watch, so the description in the figure should be uniformly vertical.

Response: Thanks for your comment! Figure 7. has been revised into uniformly vertical and the color has been changed for better understanding.

  1. Line 429. Lack of content. There should also be content after the word “and”.

Response: Thanks for your comment! It changed into a “comma and”.

  1. The discussion is not a redescription of the results and lacks logic. The title of the authors is Modulation of the Gut Microbiota with Prebiotics and Antimicrobial Agents from Pleurotus ostreatus Mushroom, while it is not reflected in the discussion.

Response: Thanks for your comment! The discussion section was extended with 2 more questions regarding the most important antimicrobial and prebiotic bioactive compounds. The discussion and conclusion were extended and clarified.

  1. The reference should be updated in recent years.

Response: Thanks for your comment! It was extended with more references in recent years.

Many thanks for your time, hoping our correction will be accepted, thanks!

Reviewer 3 Report

Dear Authors!

In the manuscript are too many repeating paragraphs with a too general description, especially in the Introduction section. The Methodology of the review and methods for selecting articles for the review manuscript are very poorly described.

Author Response

Dear Reviewer 3#

Many thanks for your time and efforts to improve our MS to be ready for publication!

In the manuscript are too many repeating paragraphs with a too general description, especially in the Introduction section. The Methodology of the review and methods for selecting articles for the review manuscript are very poorly described.

Response: Many thanks for your comment!

We tried to improve the MS to be ready for publication in the revised MS. Please check the revised MS again, which included many changes in nearly all sections of the MS! Hoping this change will meet your opinion. These changes also referred on the comments from the other two reviewers!

Concerning the part of the Methodology of the review, we found that adding a flow chart is better for the reader and improved our MS with details about the steps that can establish or create a review article. It is so easy now for the reader to follow our review and understand the steps, which we followed to write this MS.

Hoping our corrections will be accepted!

Many thanks for your time, hoping our correction will be accepted, thanks!

Round 2

Reviewer 1 Report

Dear Editors, 

The authors made all necessary changes to improve the manuscript, and now I recommend it for publication in its current form.

Reviewer 2 Report

The author has responsed the reviewer's comment point by point. It can be acceptted in current revision.